# Communicating Function and Co-Creating Healthy Food: Designing a Functional Food Product Together with Consumers

**DOI:** 10.3390/foods11070961

**Published:** 2022-03-26

**Authors:** Petar Vrgović, Milica Pojić, Nemanja Teslić, Anamarija Mandić, Aleksandra Cvetanović Kljakić, Branimir Pavlić, Alena Stupar, Mladenka Pestorić, Dubravka Škrobot, Aleksandra Mišan

**Affiliations:** 1Faculty of Technical Sciences, University of Novi Sad, Trg Dositeja Obradovića 7, 21000 Novi Sad, Serbia; 2Institute of Food Technology, University of Novi Sad, Bulevar Cara Lazara 1, 21000 Novi Sad, Serbia; milica.pojic@fins.uns.ac.rs (M.P.); nemanja.teslic@fins.uns.ac.rs (N.T.); anamarija.mandic@fins.uns.ac.rs (A.M.); alena.tomsik@fins.uns.ac.rs (A.S.); mladenka.pestoric@fins.uns.ac.rs (M.P.); dubravka.skrobot@fins.uns.ac.rs (D.Š.); aleksandra.misan@fins.uns.ac.rs (A.M.); 3Faculty of Technology, University of Novi Sad, Bulevar Cara Lazara 1, 21000 Novi Sad, Serbia; amod@uns.ac.rs (A.C.K.); bpavlic@uns.ac.rs (B.P.)

**Keywords:** co-creation, communication, functional food, NADES, raspberry

## Abstract

Functional food is lately an interesting topic from the new product development perspective; complex motivation and expectations of consumers regarding it present a challenge when new products are designed. Co-creation is an interesting alternative to the standard practices by the R & D departments since it directly involves consumers in the various stages of the creation process. This work aims to describe experiences of engaging consumers in different development stages of a functional food product within a project realized at a food research institute. Four consecutive studies were conducted: the first study explored current trends in Serbia regarding the way consumers use functional food and are informed about it; the second study described development of a raspberry seeds extract with antioxidant and anti-proliferative activity confirmed in vitro; the third study tested the same extract in a sample of consumers, validating its usability in food products; and the fourth study described a co-creation session with 18 participants, during which a number of activities were realized to stimulate idea generation. Rather than the final product idea itself, this work is valuable because of detailed insights into the various phases of the co-creation process. It is shown that consumers and food researchers can together engage in the new food product development process as long as the communication between them is rich and with mutual understanding.

## 1. Introduction

The link between diet and general health has always been visible, but it is becoming increasingly important in the new age of rising health care costs, rising living standards, and drastically changing perceptions of food; such demands may be achieved by formulating functional foods [1]. Although there is no strict definition of functional food, it can generally be said that it is food that, in addition to the basic nutritional value, has a positive effect on one or more target functions in the body [2]. Such foods include foods that are enriched with nutrients naturally present in them (by increasing the concentrations of these components) [3] or new components that are not naturally found in those foods [3,4]; further, they can be modified [3] or improved products [5].

Consumers around the world are now more than ever health conscious and more educated about what goes into their food; health is one of the most important motivators that influence consumers’ food choices [6,7], and health motivation significantly influences the way consumers perceive and process information displayed on the food package [8]. However, the relationship between consumers’ interest in their own health and the purchase of a food product is not straightforward, as different emotions arise when different food groups are mentioned to the consumers and thus mediate said relationship. In the case of functional food, although it prompts a generally positive affect with the customers, it still induces more fear compared to regular and organic food [9], and it is even documented to evoke several different emotions simultaneously from both the “positive” and “negative” categories [10].

Such a complex relationship between consumers’ intentions, underlying emotions, and attitudes towards functional food [11] implies that acceptance rates of novel functional food products may not be easily projected and could be quite worrisome [12]. It is not uncommon for a significant percentage of novel food products to fail on the market because there was a lack of response to the complex needs of consumers [13,14]. New product development (NPD) in the food industry is usually a closed, intra-organizational process with little or no contact with the end-users during the early stages—companies frequently rely on their own intuition, wisdom, or competence to come up with ideas for new products, and they become interested in consumers’ impressions only once they introduce a new product to the market [15]. Although a range of methods is available that can aid companies in generating new product ideas based on input from marketing research [16], these methods are still just indirectly voicing end-users’ perspective, as consumers usually are contacted when the company’s interest in obtaining a glimpse of the market is present or during the product-testing phase. Aiming to bridge this gap, some companies embark on a journey to create new food products together with the consumers from the start by shifting towards consumer-led product development, which is saturated with rich market communication between R&D/production departments (voice of the company) and end-consumers (voice of the consumer) [14]: the outcome of the creative process could be significantly improved if consumer requirements are integrated into the NPD objectives from the start [17]. Co-creating a new food product (NFP) together with potential end-users brings a number of advantages [18] and is recently perceived as a vital factor in NPD [19], where rich, two-way communication with the consumers in both early stages and the latter promotion stages is crucial for a product’s market success [20].

Customer co-creation—Defined as a “process in which customers consciously and actively engage in a firm’s innovation process, taking over innovation activities traditionally executed by the firm” [21] (p. 664)—Offers an interesting and effective approach to radicalize the NPD process, to humanize it [22], and to make it more spontaneous and playful [23]. In the last decade, a number of publications have appeared that describe successful cases of co-creating NFP with consumers. Kemp [24] described the case of Pepsico’s Mountain Dew^®^ “DEWmocracy™” campaign where consumers were co-creators of a new variant of a nonalcoholic beverage, with a series of crowdsourcing activities and even a story-based, interactive on-line game. Barone et al. [20] co-created healthier meat products on the basis of plant-based ingredients with lower fat and salt content. Prior to the co-creation session, consumers’ opinions, perceptions, preferences, and needs for healthier meat products were explored. Banović et al. [19] demonstrated that application of projective and creative techniques is beneficial for consumers’ creation of new ideas for aquaculture products during the early stages of NPD process, while Bettiga and Ciccullo [25] described a number of Italian food companies that co-created their products (such as gluten-free pasta, nuts biscuits, frozen pizza, and homogenized meat) by involving customers in different parts of their NPD processes.

A couple of recently published papers provide deeper insights into the factors of the co-creation process and their possible effect. Hoppe et al. [18] demonstrated that the main driver for consumer participation in innovative activities in the food industry is the innovativeness of an individual—Substantial curiosity about new food products will probably lead to significant willingness to participate in the NPD activities. Jacobsen et al. [26] examined the trust that consumers have in co-created food products and concluded that if the targeted consumer groups do not possess significant knowledge and understanding of the co-creation process, then the information about a new food product should be communicated by the consumer co-creators (their peers that were engaged in NFP development) rather than by the food company that places the product on the market. In this context, where functional foods are becoming a major focus of NPD, requiring a more open and flexible approach [27] and understanding of consumer needs (both explicit and latent) are vital for the food science and the food industry [28]; the objective of the present work was to explore and describe one research institute’s complex process of co-creating ideas for functional food products together with consumers. In the framework of a national project realized in the Republic of Serbia, which aimed to create a functional food based on an extract rich in bioactive compounds from sustainable sources by utilizing deep eutectic solvents, a multidisciplinary team of researchers engaged in a series of studies that relied heavily on rich communication with consumers and their active involvement. This team, composed of a number of scientists from the fields of food science, pharmaceutical engineering, chemistry, and psychology, designed and conducted a few interconnected studies to explore the width with which it was possible to approach the NPD process without the standard constraints of a commercial endeavor, allowing deeper insights into consumers’ needs and their creative potential.

Serbia is one of the largest world producers of raspberries (*Rubus idaeus* L.) [29], and the processing of fresh raspberries into jams, fruit pulp, juices, and other similar products generates a large quantity of seeds, which are currently underutilized in Serbia. These raspberry seeds are often discarded as waste although they contain a considerable number of bioactive compounds, such as polyphenols, tocopherols, ω-3, and ω-6 fatty acids [30]. Due to relatively high content of bioactive compounds, high availability, and low market price in Serbia, raspberry seeds were identified as a sustainable alternative source of ellagic acid and other polyphenols with antioxidant and anti-proliferative properties [31]. Ellagic acid is connected with a broad spectrum of biological effects, such as antioxidant, anti-carcinogenic, anti-obesity, anti-inflammatory and anti-angiogenic, anti-neurodegenerative, and hepatoprotective activity. In order to recover valuable compounds from industrial side-streams, sustainable extraction processes, which do not have a negative impact on the environment, have to be developed. This means, among other things, that the usage of toxic organic solvents has to be avoided at first place, but instead, the application of “green solvents”, such as Natural Deep Eutectic Solvents (NADES) [32], has to be considered.

This paper aims to describe experiences and results obtained through the complex process of creating a NFP idea based on raspberry seeds extract in order to provide evidence for embarking on an interesting journey of co-creating a functional food product together with a wide array of potential consumers, thus developing a novel and healthier food product. It contributes to the literature by identifying current trends and potential for communicating about health and functional ingredients [26,33] and also by answering the call to provide insights into the new methods and approaches to generate new ideas about food together with customers [34].

## 2. Materials and Methods

The research protocols for all studies that involved human subjects were approved by the Ethics Committee of the Institute of Food Technology in Novi Sad (Ethical approval, Ref. No. 175/I/2-3 of 3 June 2021).

### 2.1. Study 1: Consumer Preferences

The first study had an exploratory approach, utilizing a custom-made survey to collect information about consumers’ current needs, preferences, and practices related to their diet, sources of information about functional food, and ultimately, their choices and consumption of the said food. The aim of this study was to provide information that would later be used in co-creating the NFP. A concise, multi-layered survey (presented in Appendix A) was designed to assess consumers’ current needs, preferences, and practices related to their type of diet, lifestyle, sources of information about functional food, and their choices and consumption of the said food. The survey items were identified during a number of research team’s analysis sessions, where relevant publications were consulted to provide potential questions that could obtain needed information for the research project. The first version of the survey was tested on a pilot sample (*N* = 20) from general population in Serbia, to ensure all items were understandable and relevant.

By utilizing an online survey platform Google Forms to collect responses during June, July, and October of 2021, a convenient sample of 613 valid cases was obtained from the population of adult citizens from the Republic of Serbia (sample properties are presented in Table 1). Since various methods were used to promote the survey and attract potential participants (from the campaign in the university campus to different social media networks and channels), it was not possible to calculate the response rate precisely; therefore, the convenience sample used in this study should be observed with some caution regarding generalizability of results.

Statistical analysis: Excel 2016 (Microsoft Corporation, Redmond, WA, USA) and SPSS Version 26 (Armonk, NY, USA) software were used for data statistics in Study 1. Descriptive statistics, Frequencies, Crosstabs, and Multiple response commands were used to describe the sample and survey participants’ responses in order to reach the explorative goals of the survey.

### 2.2. Study 2: Consumer Ranking Test of Beverage Developed on the Basis of Raspberry Seed Extract

The second study examined results of a consumer ranking test for a soft drink made with a novel extract compared to a handful of other samples, both commercial and experimental. The intention of this study was not to speculate about the final product; rather, this study’s aim was to validate the extract regarding its potential acceptance on the market, making sure that it is likeable for use. A raspberry seed extract rich in compounds with in vitro anti-cancer activity and in vitro antioxidant activity by extraction with edible, “green” solvent (eutectic mixture consisting of citric acid, water and betaine—a novel and healthier component for food products) was developed. Details about this process have been recently published elsewhere [31]; however, for better understanding of the entire co-creation concept, some of previously published results are briefly included in the present study. In order to test the idea that produced “green” raspberry seed extract could be used as an ingredient in food products without fear that it has any off-putting properties, in the next phase of research, it was decided to create a soft drink, such as iced tea, in which the extract would serve as a carrier of bioactive components, acidifier, and colorant and to investigate how consumers like the product. 

As a result of the development process, a NADES (citric acid: betaine: water 3:1:3) extract containing 7.93 ± 0.15 mg/100 g ellagic acid was obtained. Apart from that, its in vitro antioxidant and in vitro anti-proliferative activity was confirmed [31]. In summary, raspberry seeds, side-products of raspberry processing, were used to obtain an extract rich in compounds with in vitro anti-cancer activity by extraction with edible, “green” solvent, an eutectic mixture consisting of citric acid, water and betaine. In addition to the high content of ellagic acid, the extract contains ~12.5 g betaine, which is also a bioactive compound that can serve as an osmolyte that assists in cellular water homeostasis [35]. Furthermore, due to its edible nature, the obtained NADES extract is ready to use and can have various technological roles in products, such as a source of antioxidants, acidifier (citric acid), and colorant.

Five types of similar beverage samples were included in the ranking test. Three samples were commercially available on the local market: peach-flavored iced tea (IT1), black elder-flavored iced tea (IT2), and forest fruit-flavored iced tea (IT3). The remaining two tested samples were NR and NT.

The NR sample was prepared by mixing 0.5 g of NADES (citric acid: betaine: water 3:1:3), 6 g of honey, 93.5 mL of raspberry tea, and 300 µL of blueberry fruit base. Raspberry tea was prepared with 2 tea bags, which were added to 1 L of boiling water and left for 10 min prior to further use.

The NT sample was prepared by mixing 2 g of NADES (proline–glycerol–water; molar ratio 1:2:1), 5 g of honey, 93 mL of wild thyme tea, 200 µL citron aroma, and 200 µL of pomegranate fruit base. Wild thyme tea was prepared with 5 g of plant material, which was added to 1 L of boiling water and left for 10 min prior to further use. This sample was included since the research project under which these activities were performed had an open starting point regarding the development of an extract; from the list of six relevant herbal materials, the research team narrowed down the extraction processes to raspberry and thyme. However, it will later be noticed that proline had a distinctive odor that was not pleasant for most in the consumer test.

This study was conducted in the Serbian city of Novi Sad within the 2021 “Researchers’ Night Festival”, a large and renowned cultural event in Serbia. The experiment took place in two consecutive days in one of the event facilities. A convenience sample of subjects was used: they were essentially passers-by taking part in the larger city event. In total, 93 subjects, willing to take part in testing, without aversion to the ingredients used for drink, and without any known food allergy took part in the study. Demographic information collected were gender (58 women and 35 men); age, which spanned unevenly from 18 to over 50 years old (age interval 18–30, *n* = 32; 31–40, *n* = 27; 36–50: *n* = 26; over 50, *n* = 13); and frequency of iced tea drinking (at least once a week, *n* = 21 (22.6%); at least once a month, *n* = 41 (44.1%); and never, *n* = 31 (33.3%)). The consumers were previously informed about the products and testing procedures, as described in the project approved by the Ethics Committee of the Institute of Food Technology in Novi Sad, and which referred to the survey-collaborative session. Before analysis, consumers were also instructed on the procedures describing the basic steps of the tests. Each ranking test was split for the attributes of color, odor, sweetness, sourness, flavor, and overall likeability. The lowest rank (1) corresponded to the least likeability, whereas the highest rank (5) corresponded to the most likeability. No ties were allowed; however, consumers were allowed to retaste any sample.

Samples from the refrigerator at 4 °C were distributed to consumers in 40 mL, odorless plastic bottles marked with random three-digit codes. Each consumer received five samples in a monadic sequence, with a balanced serving order (given by the XLSTAT software (Addinsoft, New York, NY, USA) to minimize systematic carry-over and position effects. For mouth rinsing, water was provided after each sample.

Statistical analysis: Excel (Microsoft Corporation, Redmond, WA, USA) and XLSTAT 2012.2.02 (Addinsoft, NY, USA) software were used for data statistics and significant difference analysis in Study 2. Rank data were analyzed using Friedman’s nonparametric test and multiple comparisons methods based on sum ranks according to ISO 8587 (ISO, 2006). Following statistically significant effects (*p* < 0.05), a post hoc LSD test was used to compare products.

### 2.3. Study 3: Co-Creating Ideas for Functional Food Product with Raspberry Seeds Extract

The third study brings the focus back to the consumers, as it compiles the results from the previous studies and, based on them, employs a co-creative approach to designing functional food products based on the developed raspberry seeds extract. In order to democratize this process and involve different stakeholders, a workshop was organized on the premises of the Institute of Food Technology, in Novi Sad, Serbia, in September 2021. This workshop aimed to utilize the creative potential of subjects who find this topic to be relevant from their perspective by including them as workshop participants and active co-creators of the NFP ideas. Evaluation of the innovative potential and emotional profile of generated product ideas was then conducted.

#### 2.3.1. The Co-Creation Session

The design of the workshop, presented in detail in Appendix B, Table A1, was inspired by Ind and Coates’s [23] understanding of the co-creation process as a spontaneous and playful activity rather than a strongly rational and controlled approach dictated by a company’s management—These authors suggest that every person can be creative enough as long as there is proper motivation and relevant knowledge about certain subject. Application of Amabile’s [36] general model of creativity helped to achieve this in the present study by influencing the three proposed components of creativity:

Motivation—During the workshop preparation, the organizers scouted for the individuals who were personally interested in food design and experimentation—The ones who saw themselves as highly curious and who observed creativity as a playful and enjoyable activity [37]; thus, this study used a convenience sample. Participants (*N* = 18) were invited individually, based on their interests and profession, to be included as one of the three groups of stakeholders: food and agricultural researchers (10 participants), food technologists and entrepreneurs in the food industry (3 participants), and end-consumers (5 participants). These participants were, during the workshop, split into four groups.

Expertise—Technical and domain-specific knowledge is necessary for successful creative activity. Ideas do not come out of nothing [38]—They are products of our past experience and new insights, where priming one’s mind with information relevant to the topic prior to the ideation session yields higher productivity and idea originality [39]. People who “brainstorm” about some topic have more success in creating quality ideas the more they know about the subject in question, so it is important to make sure they have some relevant expertise or to engage in interdisciplinary collaboration with the ones who already possess said expertise [40]. Therefore, at the start of the workshop, all participants were briefed by an expert about the purpose of the whole project but also about basic principles and benefits of using “green solvents” and then about relevant information obtained from Study 1—How the general population consumes and informs themselves about functional food and other topics, presented in Appendix B, Table A1.

Creative-thinking skills—The third component of creativity describes the importance of the way people approach problems in front of them. The more these skills are guided, improved, and trained, the better the creative output [41]; this means that, for co-creative sessions, it is not enough to gather a number of participants together, expecting them to spontaneously come about new ideas after the presentation of a challenge—Co-creation events need to be properly facilitated and guided in order to fulfil the desired goals [22]. In the present study, the workshop facilitator employed a number of techniques to stimulate participants’ creative thinking during different stages of the workshop. This first included a combination of brain purge activity [42] and the SCAMPER technique [43,44]. Second, since playing is the most important activity to unleash creative potential in children and adults alike [23], and since it is established that gamification can contribute to a pleasurable participation experience and that it can enhance task absorption [45], sets of Lego Serious Play^®^ were distributed to the participants, allowing them to fully engage in creative thinking and acting in order to develop their NFP ideas and the stories around them.

#### 2.3.2. Idea Evaluation—Originality, Feasibility, Effectiveness

The list of ideas produced during the workshop session was screened by a research associate in order to filter out entries that were incomprehensible, ambiguous, irrelevant, or redundant. The final list of ideas was then presented to a panel of three judges, two of them being experienced food technologists and researchers and the third one being a food technologist working in the food industry. All three judges were blind to the study procedure. They independently evaluated ideas on their dimensions of originality (the extent to which the idea was novel and out of the ordinary), feasibility (the extent to which the idea was precise and the ease with which it could be implemented), and effectiveness (the extent to which the idea helps to solve the problem) on separate 5-point scales ranging from 1 to 5. This approach was based on widely used Diehl and Stroebe’s idea quality-scoring method [46] for the first two dimensions, while the effectiveness dimension was added since some authors insist that it should also be observed when assessing idea quality [47]. These evaluations were then compared to determine inter-rater agreement percentages. Finally, two of the most promising ideas (the ones with the highest cumulative scores from all three judges) were selected for the next step—Evaluation of consumer emotions.

#### 2.3.3. Idea Development and Evaluation on Emotional Dimensions

The two ideas that received highest total scores from the three judges were further developed by three members of the research team using transcripts from the Lego Serious Play^®^ session and insights from Studies 1 and 2. This resulted in a concise description of these two NFP ideas, together with information about the health function of the added raspberry seeds extract (in the following part, these two ideas will have their titles marked with an “f” letter: P1f and P2f). Next, these descriptions were censored to remove information related to the raspberry seeds extract functions in order to explore if the information about function altered the consumers’ emotional response (in the following part, these two ideas will have their titles marked without an “f” letter: P1 and P2). In this way, two pairs of NFP ideas were created: the first pair with a brief information about how the product carries additional health function due to the extract being present and the second pair without the said information, mentioning the extract without its potential health benefits.

A brief list of emotions was constructed to measure emotional response from the consumers when they are presented with the two NFP ideas. This list was based on Laros and Steenkamp’s [9] hierarchical model of emotions in consumer behavior related to functional food and other types of food, which was built upon the previously established Richins’s Consumption Emotion Set [48]; only the list of “basic emotions” that Laros and Steenkamp proposed was used. However, since the same authors suggested that their list should not be taken for granted, advising other researchers to modify the list if they believe there is a good reason to do so, it was decided to enrich this list with two additional emotions that Barrena et al. [10] observed when measuring consumers’ emotional response regarding functional food surprise and worry.

An online survey was then designed to measure consumers’ emotional reactions to the NFP idea descriptions, using Google Forms. After a short description of the research project and the participants’ consent form, this survey asked participants for a few basic pieces of information about themselves. Study participants were next randomized into one of two groups: group 1 received information about additional health function, while in group 2, information about additional health function was omitted. On the next survey page, participants were presented with a short description of the first NFP idea, followed by the following instruction:

“Read carefully this description of a food product. Think for a moment about how that description makes you feel. After that, mark on a scale from 1 to 5 how much you currently feel each of these feelings, in relation to your reading of the product description. (Mark 1 means ‘I don’t feel this emotion at all’, Mark 5 means ‘I feel this emotion very strongly’). Then move on to the next question.”

The order of emotions presented for evaluation for each food product idea was randomized for each participant in order to reduce potential bias. Next, participants were presented with the next survey page, which provided a description of the second NFP idea, followed by the same instruction and the task with a list of ten emotions, again randomized. After this, the survey ended, and the participants were thanked for their contribution. In this way, for each of the two NFP ideas, two groups of participants were established: the ones who expressed felt emotions after reading product description that contained information about health function of the raspberry seeds extract and the ones who expressed felt emotions after reading product description that did not contain information about the extract’s health function. This survey was conducted in three days during February 2022, and since various methods were used in parallel to promote the survey and attract potential participants, it was not possible to calculate the response rate precisely.

Statistical analysis: Excel 2016 (Microsoft Corporation, Redmond, WA, USA) and SPSS Version 26 (Armonk, NY, USA) software were used for data statistics in Study 3. Descriptive statistics were used as well as Mann–Whitney U test due to the data not being normally distributed.

## 3. Results and Discussion

### 3.1. Study 1: Consumer Preferences

Participants from the sample first described their general health and their diet. A total of 70.1% of the participants believe that their diet is average regarding its health status, 12.6% believe that their diet is not very healthy, while 15.2% describe their diet as very healthy; 2.1% did not answer this question. Regarding the diet regime, 90.4% of the participants describe their diet as conventional in generally consuming all food groups, 2.6% stated that their diet is vegetarian, 1.0% as vegan, and 5% reported that their diet is special due to a medical condition or their personal decision. Dietary supplements are regularly consumed by 37.2% of the people from the sample, and these supplements are mostly consumed in the form of pills and capsules (69.7%), then powders (12.3%), ready-made drinks (8.3%), drops (6.1%), and syrups (3.5%).

Observing the self-reported frequency of functional food consumption (appropriate description was written in the survey, ensuring that study participants understood this term properly), the following percentages were calculated for each of the suggested answers: never 17.8%, seasonal 11.8%, several times a month 22.2%, several times a week 29.6%, and daily 18.6%. Out of those who consume functional food at least several times during one month, 35.3% do not care whether the food product is of domestic origin, 26.9% prefer domestic producers, while 37.3% are more interested in the reputation of the manufacturer and the product itself.

Participants who consume functional food several times a week or on daily basis had option to rate different aspects of functional food packages regarding the importance for their decision for purchase on a Likert 1–5 scale: this set of questions was answered by 162 participants (26.4% of the whole sample), and the highest score was recorded for the “description of the beneficial effect” (mean μ = 3.42, standard deviation σ = 1.261), next for the “description of ingredients” (μ = 3.27, σ = 1.230), followed by “package design” (μ = 3.08, σ = 1.191), while the least important aspect of the food package was the “name of the product” (μ = 2.63, σ = 1.228). When asked how they inform themselves about the functions of food/diet products (multiple answers could be marked, hence the percentages surpass 100 in sum), the respondents stated that they mostly use packaging/declaration of food products (53.7% of cases) and internet portals (53.4%); all other sources of information are less frequently utilized (Figure 1).

When asked how they inform themselves about new products (food supplements and functional foods) on the market (multiple answers could be marked, hence the percentages surpass 100 in sum), social networks were the dominant channel of communication (45.5%), and relatives and friends were also sources of news regarding this topic (24.6%), while store staff (23.4%) all other sources of information were less frequently utilized (Figure 2). If the respondents consume food supplements frequently (which was reported by 29.2% of the study participants), the most dominant reasons were (multiple answers could be marked, hence the percentages surpass 100 in sum) preserving health and improving immunity (59.5%) and improving nutrition (46.5%), followed by other reasons presented in Figure 3. Respondents prefer to consume these products mostly in a form that they can prepare themselves (effervescent tablets, tea, instant drinks, etc.) (30.1%) and pills and capsules (26.5%), followed by already-prepared beverages (juice, water, bottled tea, shot, etc.) (14.3%), powdered or grainy or mushy meals (11.0%), powders (7.4%), bars (7.4%), and drops (3.3%).

Study participants observed scientific evidence of efficiency as the most important factor when choosing a functional food product (28.9%). Composition is also important (21.7%), while taste (11.8%) and other people’s recommendations (7.7%) are not so important for those who do consume functional food products. The most important findings from this study were compiled into a concise report to be presented orally to the Study 3 participants.

### 3.2. Study 2: Consumer Ranking Test of Beverage Developed on the Basis of Raspberry Seed Extract

Due to the simplicity of the procedure, the ranking test as a traditionally discriminative test was used in a number of consumer studies [49,50,51,52]. Therefore, the test applied in this research aimed at gaining the first insight into consumer preferences related to analyzed beverage samples. Moreover, the obtained results provided us with the information on how attractive consumers found two newly created NADES products (NT and NR) in relation to other similar iced tea products on the market. Figure 4 shows the results of the ranking test performed on five beverage samples. For all ranked attributes, significant (*p* < 0.05) differences were evidenced among five samples, indicating that consumers’ preferences contributed to obtaining differences between the ranked samples.

For the color, it can be clearly seen that NR was the best ranked compared to the remaining test samples. At the same time, the color of NT products was the least acceptable in relation to the samples of iced tea samples from the market. Furthermore, consumers showed a smaller difference in likeability of NT sample for odor property compared to IT1 as the best ranked sample as well as for sourness of NR sample compared to IT3 and IT1, the best ranked samples. According to the results (Figure 4), consumers demonstrated better preferences for IT1 and IT3 samples, indicating that the preference between them differed significantly in relation to sweetness, flavor, and overall likeability. However, results obtained for NR are promising since this sample was superior to other commercial samples in terms of color, while it was comparable to IT2 in terms of overall likeability, sweetness, and flavor property.

### 3.3. Study 3: Co-Creating Ideas for Functional Food Product with Raspberry Seeds Extract

During the workshop, participants individually and in groups produced a total of 46 starting ideas related to raspberry seeds extract as a component in a food product—Many of these ideas were modifications or improvisations based on ideas previously suggested by other participants, which was expected with the usage of brain purge and SCAMPER ideation techniques. Upon initial inspection, it was possible to categorize generated ideas into one of the following groups based on the form of the final product with the extract:
Soft drinks such as iced tea, energy drink, or hot chocolate beverage;A component for application in producing sweet pastries and cakes;Sauce or a marinade for meat or fish;A final food product that is ready and sufficient on its own for consumption.

After eliminating ideas that obviously lacked novelty (market is already saturated with similar products) and after merging similar and redundant ideas, the final list of 15 ideas was compiled. This list was then presented to the three independent judges for originality, feasibility, and effectiveness evaluation. Defining the three judges’ ratings as being in agreement whenever their evaluations fell within one point of each other (for example, agreement was recorded if an idea received marks 4, 4, and 3, and disagreement was recorded if an idea received marks such as 5, 4, and 3) [46,53], it was found that all three judges were in agreement in 6 out of 15 ideas regarding the originality evaluation, in 9 out of 15 ideas regarding the feasibility evaluation, and again in 9 out of 15 ideas regarding the effectiveness evaluations. More generally, if it was counted that all three judges had agreement on at least two out of the three idea-quality dimensions regarding every idea, their evaluations were marked as an agreement for 9 out of 15 ideas. Results of judges’ evaluations on three idea-quality dimensions are presented in Table 2, with their detailed evaluations in Appendix C, Table A2.

Ideas number 11 (vegan soy sausage with raspberry seeds extract) and 5 (rice pudding with raspberry seeds extract) received the highest number of points on all three dimensions of idea quality combined, and since both of them had satisfying agreement between the judges, these two ideas were chosen for further development and analysis of emotional response. As described earlier, they were further developed by the research team to produce concise descriptions of a NFP idea. These descriptions were then formulated with (containing letter “f’ at the title code) and without (not containing letter “f” at the title code) information about functional descriptions for the raspberry seeds extract, resulting in the following four descriptions:
(1)P1f Vegan soy sausage with raspberry seeds extract

This product does not contain raw materials of animal origin and is suitable for vegetarians, vegans, and fasting days. The product is made of soy and does not contain artificial preservatives, colors, and flavors. It is enriched with natural raspberry seed extract that contains ellagic acid, a compound with potential anti-cancer activity and antioxidant activity. It also contains betaine, which lowers high levels of homocysteine in the body, which contributes to reducing the risk of cardiovascular, cerebral, peripheral vascular, and neurodegenerative diseases. It has a pleasant, slightly salty taste and is designed to imitate several sensory properties of animal meat, such as appearance, taste, texture, and mild pink color, which makes it a delicious alternative to meat products. The product is a rich source of protein. It does not contain GMOs.
(2)P1 Vegan soy sausage with raspberry seeds extract

This product does not contain raw materials of animal origin and is suitable for vegetarians, vegans, and fasting days. The product is made of soy and does not contain artificial preservatives, colors, and flavors. It is enriched with natural raspberry seed extract. It has a pleasant, slightly salty taste and is designed to imitate several sensory properties of animal meat, such as appearance, taste, texture, and mild pink color, which makes it a delicious alternative to meat products. The product is a rich source of protein. It does not contain GMOs.
(3)P2f Rice pudding with raspberry seeds extract and vegetable milk

This product does not contain raw materials of animal origin and is suitable for vegetarians, vegans, and fasting days. The product is made from rice and vegetable milk and does not contain gluten, lactose, milk protein, or artificial preservatives, colors, and flavors. It is enriched with natural raspberry seed extract that contains ellagic acid, a compound with potential anti-cancer activity and antioxidant activity. It also contains betaine, which lowers high levels of homocysteine in the body, which contributes to reducing the risk of cardiovascular, cerebral, peripheral vascular, and neurodegenerative diseases. It has a pleasant, slightly sweet taste and a slightly pink color.
(4)P2 Rice pudding with raspberry seeds extract and vegetable milk

This product does not contain raw materials of animal origin and is suitable for vegetarians, vegans, and fasting days. The product is made from rice and vegetable milk and does not contain gluten, lactose, milk protein, or artificial preservatives, colors, and flavors. It is enriched with natural raspberry seed extract. It has a pleasant, slightly sweet taste and a slightly pink color.

The online survey about consumers’ emotional reactions to NFP ideas was successfully completed by 126 adult participants; age span was from 18 to 70, with a mean of 40.74; 88 participants were female (69.8 percent). Regarding consumption of functional food, 33 participants (26.2%) stated that they never consume it, 76 participants (60.3%) stated that they consume it to a smaller extent, and 17 participants (13.5%) to a large extent. Regarding consumption of dietary supplements, 37 participants (29.4%) stated that they never consume them, 46 participants (36.5%) stated that they consume them to a smaller extent, and 43 participants (34.1%) to a large extent. Most of the participants, i.e., 107 (84.9%), described their diet as conventional; 13 participants (10.3%) described their diet as vegetarian or vegan; and 6 participants (4.8%) stated that they have a special diet due to health concerns or personal choice.

This survey showed that study participants feel emotions with a positive affect, as defined by Laros and Steenkamp (contentment, happiness, and love) [9] to a larger extent in the case of rice pudding with raspberry seeds extract than for the vegan soy sausage with raspberry seeds extract (Figure 5). This finding is expected since rice pudding is a common dessert in Serbian households (as elsewhere) although usually prepared with cow’s milk and only sometimes with addition of fruit extracts. Rice pudding, thanks to its mild taste and high nutritional value, is an option frequently offered to small children and, at least in Serbia, a common dessert allowed for consumption when a child is having stomach problems. These facts surely cause this product to be associated with a set of pleasant emotions, such as relief and comfort, which may lead to the emotion of contentment and even love, making it a product with sentimental value [9] for many. Happiness is also an expected reaction regarding the rice pudding, as sweet taste has a metaphoric association with this emotion [54]. Since vegetable milk is a relatively a close alternative to cow’s milk, and since fruit extracts similar to the raspberry seeds extract are often used in various desserts in the Serbian cuisine, this NFP idea was expectedly followed with pleasant emotions, as described. 

The same cannot be said for the vegan sausage, which is surely not perceived as a known, sweet, or comforting product by most consumers in Serbia. The description of vegan soy sausage with raspberry seeds extract evoked emotions with a negative affect to a larger extent (sadness, fear, anger) probably because the sausage as a form of food is mostly associated with pork meat in Serbia. The pork sausage is one of the most frequently used meat products in Serbia, and different parts of the country have their own regional varieties [55,56], making this product a part of national heritage and pride for many. The notion of a vegan product being associated with a name of a food product that is usually based on meat may thus bring unpleasant emotions and negative views from an average consumer [57] at least in Serbia, where vegan food is not consumed by many. This may also explain the significantly higher level of worry felt while reading about vegan sausage, as it introduces soy filling to a product shape that is usually filled with spicy meat. An additional reason for the elevated level of this emotion may be found in the fact that raspberry seeds extract is added in the already controversial product, where raspberries are almost exclusively perceived as something utilized in sweet food products on the Serbian market. These unpleasant emotions may additionally be exacerbated by the situation in which the raspberry is considered by many as one of the most recognizable Serbian food products but never associated with savory food or in combination with the word sausage. Emotions of pride and shame did not differ between the two product ideas (Figure 5); explanation for this may lie in the fact that the emotion of pride generally occurs when a consumer feels superior compared to another person regarding some achievement or a property [9], while the emotion of shame occurs when an individual perceives a failure of some aspect of the global self [58]—None of which apply here since the NFP ideas were not generated by the emotional evaluation study participants. 

One additional observation may be interesting. After the survey completion, the research team received a few informal comments by some study participants regarding their interest in the study and the way it was conducted. One interesting comment by a handful of participants stood out: they asked why the emotion of “disgust” was not present as an option, as they stated that they were “disgusted by the idea of a vegan soy sausage with raspberry seeds extract”. While we understand the skepticism of some regarding a food product that may sound controversial in a country where pork products are a part of national identity, while vegan products are not widely consumed and frowned upon by many, we provide the following explanation on why the emotion of disgust was not included in this study, in addition to the methodological argument that the studies that we relied on ignored the said emotion [9,10,48]. Shaver provided detailed evidence on basic human emotions and commented that, in the modern age, the emotion of disgust has shifted away from a primary physical and emotional reaction (with distinct facial expression and supposed links to innate reactions to bad tastes and smells) to a type of anger akin to contempt [59] (pp. 1069). In other words, the word “disgust” in an adult’s emotional lexicon has dominantly transformed from an emotional reaction into an evaluative statement that aims to disprove the object of disgust.

Interestingly, for both products, there were no significant differences found regarding the intensity of emotions felt in relation to whether the information about functional effect of the raspberry seeds extract was disclosed or not. Only when the levels of emotions were observed, regardless of the food product (for both products together), a significant difference was found on one dimension, contentment, showing that participants who did not receive information about functional effect felt more content than the ones who did receive the said information (mean ranks 137.15 and 117.98, respectively, with Mann–Whitney U value of 6647.000, Z = −2;144, *p* = 0.032). A possible explanation could be that mentions of various diseases (cancer, cardiovascular, cerebral, peripheral vascular, and neurodegenerative diseases) as part of the information about functional properties of the extract upset the readers, thus making them feel less contempt. Finding that information about functional effect does not generally affect emotional response is somewhat contradictory to the finding from Study 1, where it was found that people who regularly consume functional food rated “description of the beneficial effect” as the most important factor for their decision to buy functional food. A possible explanation would be that Study 4 did not include motivational aspects (cognitive and conative) of functional food consumption, asking participants to only rate their emotions regardless of their current needs for a food with additional function. 

## 4. Conclusions

This work, complex in its approach, aimed to explore and describe a food research institute’s process of co-creating a functional food product together with a wide array of potential consumers. It contributes to the contemporary literature by providing an overview of methods and actions for co-creation of NFP from the perspective of a research institute as the main actor rather than a commercial company. Although the current literature provides examples of NFP co-creation in the food industry, there are not any records available regarding these activities in a non-commercial context. The present work documents various activities that could be also utilized by other organizations elsewhere when the creative potential of consumers, researchers, and producers is combined. Acknowledging the importance of communication during the NFP co-creation [26], it was this multidisciplinary team’s intention to engage in action research together with relevant and diverse stakeholders in order to revalorize raspberry seeds as a source of nutrients by utilizing natural compounds and to produce more than just an extract—to produce an idea for a healthy, functional food product.

The first study described current needs of consumers in Serbia regarding functional food. It was found that consumers do use functional food products to a good extent, mostly with intention to preserve their health but also to improve their diet or for reasons related to sports or health issues. For a number of study participants, information about functional food and about scientific evidence of efficiency is quite important—product information is especially important for dietary considerations, and these are probably conscious food shoppers [60]. Social networks and personal communication with friends and relatives are the main channels for information about new functional food products, which is in line with previous research that suggested a more personal approach when advertising NFP [26]. In the second study, the NADES raspberry seeds extract showed satisfactory ratings in the consumers ranking test, qualifying it for usage in the next step. Finally, the third study depicted activities related to next stages of the NFP design process—instead of R&D departments or food technology specialists being responsible for this step, it was decided to gather a number of interested stakeholders and to explore potential usages for the raspberry seeds extract together with them. Through a playful set of activities, a number of interesting NFP ideas emerged; an evaluation panel selected two of these ideas as the most promising, and these two ideas were further developed into NFP descriptions subject to an evaluation of emotional reactions by a sample of the general public in Serbia. The idea of a rice pudding with raspberry seeds extract and vegetable milk showed most potential and evoked mostly pleasant emotions within a sample of consumers, which was explained with similarities of the proposed dish with a dessert that is cherished in Serbia as well as around the world, being more acceptable to an average consumer than the other food product idea, the vegan sausage.

A number of limitations, such as convenience samples in three studies from just one country, relatively small samples in Studies 2 and 3, a limited set of resources, and just one co-creation session as well as relatively modest input from the industry, are definitely present. However, aiming to show that open communication and creativity stimulation have a significant role in NFP development, this work will hopefully inspire other researchers and companies to create a stake in co-creating functional food products together with their consumers. This research approach was, surely, far more explorative than confirmative in its nature. The study presents experiences of a research institute’s project to design novel food within a research project. Therefore, the results of our studies can hardly be compared to the traditional R&D practices, and it would be troublesome to benchmark the practicalities and benefits of this approach, as it is not commercial in nature. The final limitation of this work is that the final NFP idea was not validated in terms of actual sensory quality or consumer acceptance, the reason for which lies in the fact that the aim of the project was limited to the development of a functional extract and its potential usage, without activities related to final product validation.

## Figures and Tables

**Figure 1 foods-11-00961-f001:**
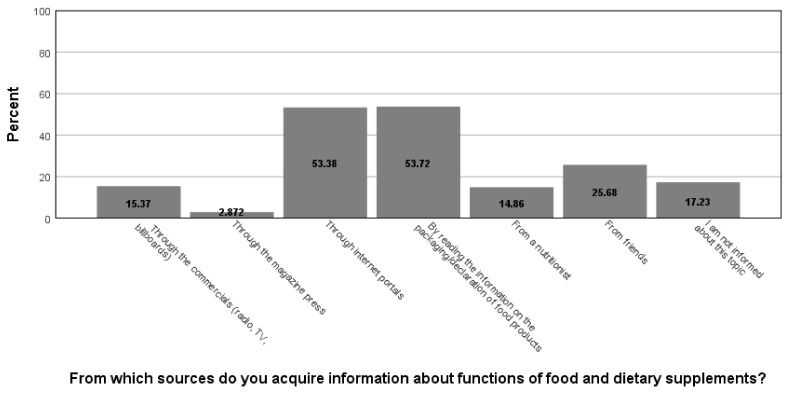
Sources of information about functions of food and dietary supplements.

**Figure 2 foods-11-00961-f002:**
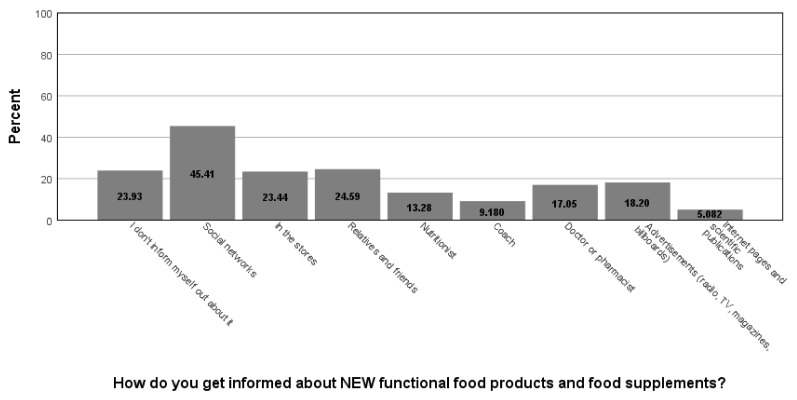
Sources of information about new functional food products and food supplements.

**Figure 3 foods-11-00961-f003:**
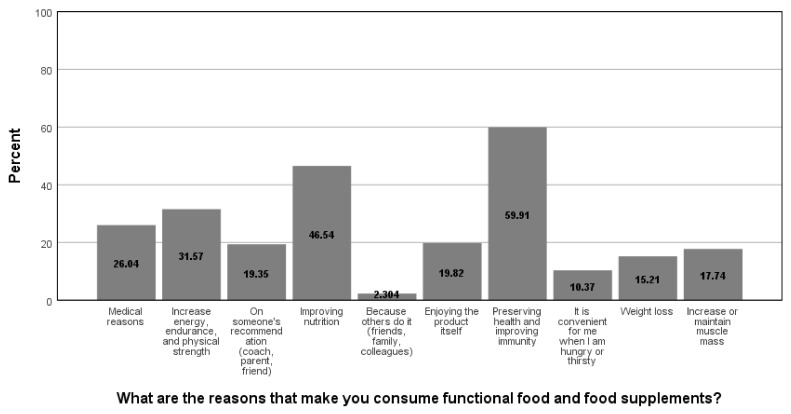
Reasons for functional food and food supplements consumption.

**Figure 4 foods-11-00961-f004:**
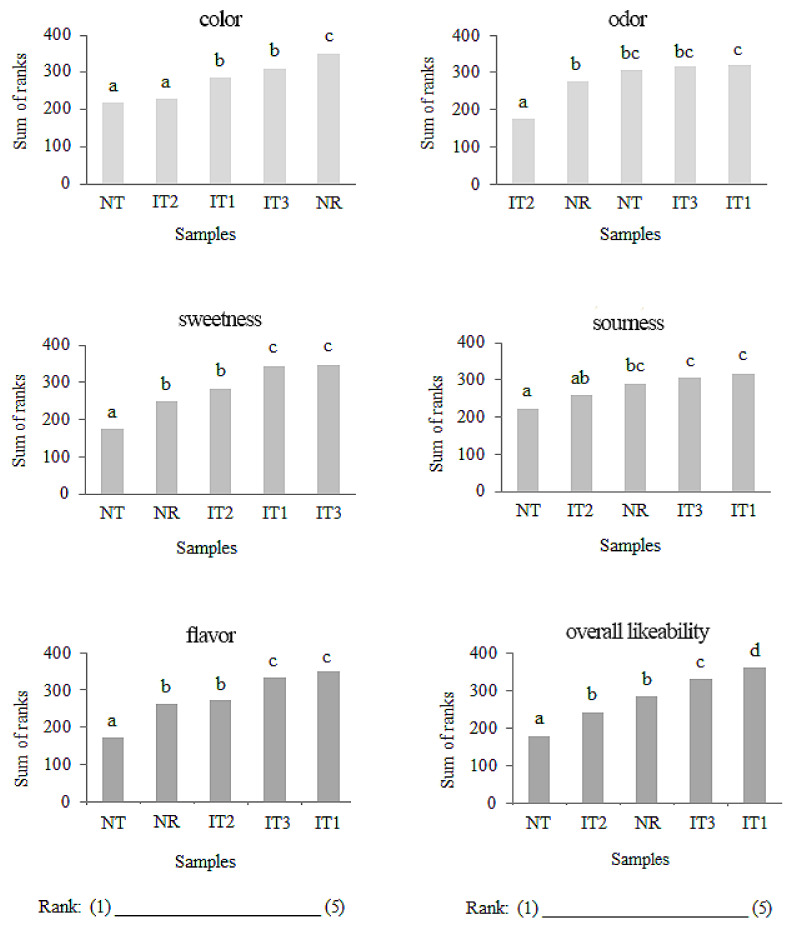
Results of preference ranking test of beverage samples. IT1, peach-flavored iced tea; IT2, black elder-flavored iced tea; IT3, forest fruit-flavored iced tea; NT, NADES thyme extract-based soft drink; NR, NADES raspberry seeds extract-based soft drink. Values marked with the same letter are not statistically different (*p* < 0.05).

**Figure 5 foods-11-00961-f005:**
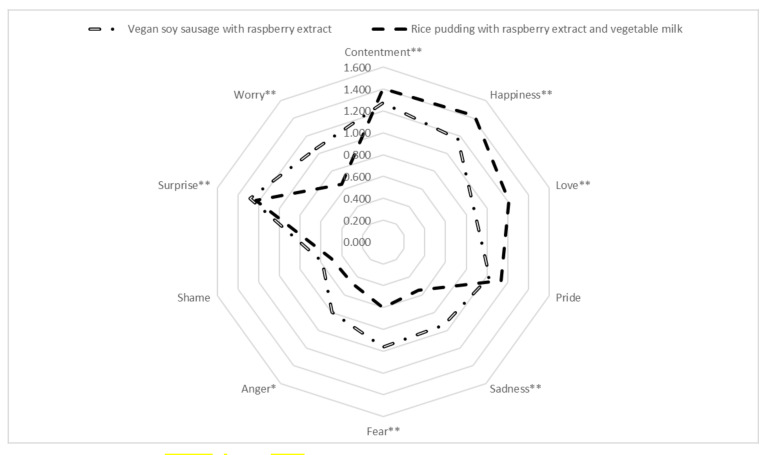
Radar chart of emotions felt while reading about the two product ideas. Emotions marked with * showed differences significant at 0.05 level; those marked with ** showed differences significant at 0.01 level.

**Table 1 foods-11-00961-t001:** Sample properties from Study 1.

Sample Property					
Gender	Female		Male		
	71.10%		28.90%		
Age	Minimum	Median		Maximum	
	18	32		76	
Working status	Permanently employed	University students	Occasionally employed	Unemployed	Retired
	38%	34.6%	9.6%	7.3%	3.6%

**Table 2 foods-11-00961-t002:** Final list of NFP ideas and sums of the judges’ evaluation points. Judges’ agreement was recorded when at least two of the three idea-quality dimensions were identical.

Idea	Judges Had Agreement	Sum for the Originality Points	Sum for the Feasibility Points	Sum for the Effectiveness Points	Total Sum of Points
1. Fruit yogurt with raspberry seeds extract. Ingredient for fruit cake or ice cream, not for direct consumption	No	11	12	12	35
2. Ice cream with raspberry seeds extract	Agreement	9	15	10	34
3. Butter with raspberry seeds extract for cake and pastry	No	13	11	11	35
4. Instant ice cream with raspberry seeds extract	Agreement	8	11	11	30
5. Rice pudding with raspberry seeds extract	Agreement	11	15	12	38
6. Healthy jelly cube-jelly honey with raspberry seeds extract	No	13	9	9	31
7. Sweet noodles with raspberry seeds extract	No	12	10	9	31
8. Meat sausage with the addition of raspberry seeds extract	No	11	8	8	27
9. Meat dressing with raspberry seeds extract for roasted meat	Agreement	11	13	10	34
10. Marinade for meat and fish with raspberry seeds extract, before cooking	No	8	9	9	26
11. Vegan soy sausage with raspberry seeds extract	Agreement	14	13	12	39
12. Hot chocolate drink with raspberry seeds extract	Agreement	10	14	11	35
13. Topping for cake, pancake, or ice cream with raspberry seeds extract	Agreement	9	15	12	36
14. Raspberry seeds extract as a liquid additive for sweet dough and cake	Agreement	7	11	8	26
15. Stuffing for muffin and donut with raspberry seeds extract	Agreement	8	14	12	34

## Data Availability

Not applicable.

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
