# Peer review of "Communicating Function and Co-Creating Healthy Food: Designing a Functional Food Product Together with Consumers"

_foods, 2022, doi:10.3390/foods11070961_

Round 1
Reviewer 1 Report
The paper describes a methodology for developing new products involving consumers in the initial steps of the process. The study is interesting, but it must be rewritten to ease understanding and comprehension. I encourage authors to present one introduction section without subdivision, one material and methods section, and one results and discussion section.
INTRODUCTION
It will be convenient to present a complete introduction section without subsections. I suggest authors to include more information on sustainability, bioactive compounds, etc. Also, a redefinition of the structure of the introduction will help. A suggestion could be:
- Fuctional foods
- Consumers needs, choices and expectations
- Co-creation methods
Information compiled in section 1.1. should be moved before line 48.
Lines 90 to 103: this is information related with the project aim and consortium. From my point of view, this is not necessary to be included because of confusing the reader.
Lines 103 to 123 should be moved to material and methods section.
Section 1.2 could be integrated in lines 60 to 80.
Lines 195 to 221 explained different studies in which co-creation has been used. It is not necessary to explain all this information, one or two examples should be enough.
The aim of the paper must be defined at the end of the introduction section.
MATERIAL AND METHODS
At the beginning of this section an explanation of the different developed studies should be presented (information included in lines 103 to 123).
STUDY 1 - I recommend authors to sum up the sociodemographic information of lines 249 to 257 in a table.
STUDY 2 – Lines 326 to 339 belongs to introduction section.
STUDY 3 – Lines 390 to 398 should be included at the beginning of mat and methods.
STUDY 4 - Lines 484 to 489 should be included at the beginning of mat and methods. Also, explanation about co-creation must be reduced.
RESULTS
I suggest the authors present the results of study 1 using figures to facilitate understanding.
Reviewer 2 Report
This manuscript describes the combination of four separate studies into a co-creation process for a functional food product. However, empirical evidence of the practicality or benefits of this complex process compared to traditional R&D practices is not presented. No data is provided to validate the final product concepts in terms of actual sensory quality or consumer acceptance of such products. The study does provide interesting data on Serbian consumers' decision making and emotions related to functional food products. Authors, please also find specific questions and concerns below.
1. Line 50: Please replace the two citations with more contemporary support of the claims about current motivators of consumer food choice.
2. Perhaps consider curtailing the three-and-a-half page Introduction to a more concise section.
3. It is unclear what this work contributes to the science of co-creation that is lacking from other literature cited in the manuscript.
4. If "Study 2 Raspberry seeds extract development" of this manuscript is previously published research from reference [42], please explicitly indicate this. If this is the case, Study 2 should not be presented as new original research.
3. In "Study 3 Raspberry seeds extract consumer ranking test," it is unclear why the "green" thyme extract soft drink was included.
4. Did the authors measure absolute liking or acceptability of the NR sample and other beverages?
5. Lines 468-471 and 481-482: The presently discussed consumer preference test was affective in nature, while trained descriptive sensory panels yield analytical sensory data. One should not be substituted for another, as they provide different types of information.
